# The "Angel of Light at Work": An Assessment of the Christian Mission in the Southern Hemisphere

Raphael Okitafumba Lokola

Faculté de Psychologie et Sciences de l'Education, Université de Lodja, Ville de Lodja B.P. 155, Democratic Republic of the Congo; raphlokola@gmail.com

**Abstract:** Today, it is commonplace to hear that Africa is the hope of Christianity. Using the Ignatian image of "the angel of light", this paper proposes to qualify this belief. I will show that this belief is verified quantitively. However, the lack of authentic Christian witness makes the above assertion problematic. My analysis will focus on the tradition of the *églises de réveil* (awakening churches) in Francophone Africa, especially in the Democratic Republic of the Congo. This contemporary tradition is a hybrid Christian movement that combines Evangelical and Pentecostal attitudes. I will develop my analysis in three steps. First, I will discuss the nature of *églises de réveil*. Second, I will describe the modus operandi of these awakening churches. My discussion will indicate how this modus operandi works in urban and rural areas. Lastly, I will propose one perspective for authentic Christian growth. I will argue that a Christian mission needs to be radically Christocentric. This radical Christocentrism revolves around two major axes, namely a sign of contradiction vis-à-vis the world and a radical witness of the Gospel of Jesus Christ.

**Keywords:** Eglises de réveil (awakening churches); Christocentrism; sign of contradiction; Christian witness; Democratic Republic of the Congo; angel of the light





## 1. Introduction

Today, it is common to hear that Africa is the hope of Christianity. For example, in its April 2023 report, Statistica pointed out that "Christianity is the major religion in numerous African countries. As of 2023, around 96 percent of the population of Zambia [is] Christian, representing the highest percentage on the continent. Seychelles and Rwanda [follow] with roughly 95 percent and 94 percent of the population being Christian, respectively. While these countries present the highest percentages, Christianity [is] also prevalent in many other African nations." (Galal 2023) Among those African countries where over 85% of residents adhere to Christianity, the exact proportions are as follows: the Democratic Republic of the Congo (DRC), 92%; the Republic of the Congo, 90.7%; Namibia, 90%; Lesotho, 90%; Cape Verde, 89.1%; Equatorial Guinea, 88.7%; Uganda, 88.6%; Gabon, 88%; and Zimbabwe, 87%.[1] Similarly, J. J. Carney notes that "If Africa, and to a lesser extent Asia, have seen remarkable growth, Europe has experienced a massive declension in Catholic baptisms, marriages, eucharistic practice, and membership." (Carney 2022, p. 27).

These accounts give a good idea of the quantitative growth of the Christian faith on African soil. Despite their relevance, I am more interested in qualitatively assessing the Christian faith in Africa.

From this standpoint, using the Ignatian image of "the angel of light", this paper[2] proposes to qualify the belief according to which Africa is the hope of Christianity. As mentioned above, this belief is quantitively verified. However, the ensuing analysis will show that the lack of authentic Christian witnesses makes the preceding assertion problematic.[3] My analysis focuses on the tradition of the *églises de réveil* (awakening churches) in Africa, especially in the Democratic Republic of the Congo (DRC). This choice is not arbitrary. Previously, my research appraised the Christian faith in the Roman Catholic tradition.[4] In

addition, considering Laurent Larcher's statement that "Protestantism is the main religious current on the continent [of Africa]", (Larcher 2018, pp. 22–23) I believe that an objective assessment of this Christian tradition is apropos.

It is worth noting that my analysis focuses on the Pentecostal experience in Francophone Africa, as that is often overlooked.[5] I will develop this analysis in three steps. Firstly, I discuss the nature of *églises de réveil*. Secondly, I describe the *modus operandi* of these *églises de réveil*. My discussion indicates how this *modus operandi* works in urban and rural areas. Lastly, I propose a measure for determining authentic Christian growth. I will argue that a Christian mission needs to be radically Christocentric. This radical Christocentrism revolves around two major axes, namely a sign of contradiction vis-à-vis the world and a radical witness of the Gospel of Jesus Christ.[6]

## 2. The Nature of *Eglises de Réveil*

Bluntly put, the *églises de réveil* (awakening churches) are the outgrowth of the Evangelical and Pentecostal churches. They are, thus, hybrid Christian movements that combine Evangelical and Pentecostal attitudes. Why do these churches bear this name?

René Girault and Jean Vernette point out that "No religious group likes to be called a 'sect' because of the pejorative character that connotes the adjective '*Sectaire*' in French." (Girault and Vernette 1979, p. 473) In this language, *Sectaire* is synonymous with being fanatical or intolerant. This fact may explain the avoidance of the word "sect" and the choice of the awakening church by contemporary Christian movements.

Taken at face value, an awakening church is reminiscent of the Psalmist who says, "My heart is steadfast, God, my heart is steadfast. I will sing and chant praise. Awake, my soul; awake, lyre and harp! I will wake the dawn. I will praise you among the peoples, Lord; I will chant your praise among the nations." (Psalm 57: 8–10). From the perspective of the members of these churches, the name "awakening church" conveys the idea that the outpouring of the Holy Spirit has awakened them. Their attitude responds to Saint Paul's injunction: "It is the hour now for you to awake from sleep." (Romans 13: 11). They are supposedly filled up with the gifts and fire of the Holy Spirit like on Pentecost day (Acts 2: 1–41) to bear witness to Jesus Christ. That is why long hours of ear-piercing praise and worship characterize their prayer gatherings.

In addition, the awakening church can represent renewal or revival. Girault and Vernette show that "the questioning of the order in place can only result from a *new* conception of the divine order, generally elaborated by a man who declares himself directly inspired by the [Holy] Spirit." (Girault and Vernette 1979, p. 474). From this perspective, each revival church revolves around its pastor. As Girault and Vernette point out, "the influence of the prophet on the group is due in particular to his charismatic character. He appears as a man gifted with strengths and qualities above the ordinary to which his followers attribute a supernatural origin: he is the envoy of God." (Girault and Vernette 1979, p. 475). This pastor must, thus, prove that he is the envoy of God by his exceptional eloquence as a preacher and his power to operate miracles, especially spectacular healings.[7] My analysis will unveil that these expectations have led many pastors to drift to the use of non-evangelical means in their ministries.

For the time being, I agree with Girault and Vernette that the *églises de réveil* constitute religious movements whose goal is to shake off the torpor and lukewarmness of existing churches. These *églises de réveil* are, therefore, "numerous and worthy attempts to rediscover a purer and more evangelical Christianity." (Girault and Vernette 1979, p. 479). Similarly, Robert Shako Lokeso gives insight into the complex nature of these churches. In his book on the Church as the sign of salvation, he writes, "Given these relevant analyzes and observations, it emerges that sects constitute a protest against the religious system in place and against the social, political, economic, judicial, interpersonal and other order."[8] He also clarifies why the sects protest against the established religious and social order as follows:

> "In short, if the sects and the new churches in Congo-Kinshasa can be considered
> as a questioning of the established social and religious order; if they constitute

a warning signal and a challenge to political-administrative leaders and those of institutional churches to review the social advancement of man; if they show the limits of our rites, our liturgies, our preaching and our religious practices; they do not however satisfy the deep aspiration of the Congolese. The deviations noted are signs of concern. They pose a great challenge not only to the Churches but also and above all to the survival of the nation." (Shako Lokeso 2018, p. 66)

Shako Lokeso suggests that these sects, including the awakening churches, stand as an alternative order that prioritizes addressing the population's existential issues. This analysis does not vindicate the approaches of these awakening churches.

All things considered, I assert that it is difficult to infallibly pinpoint the nature of the awakening churches, as they recurrently sprout like mushrooms and die like flies. Despite their complexity and indeterminate diversity, two accounts help to identify some key features that make comprehensible the nature of the modern awakening churches.

From the perspective of Mélanie Soiron Fallut, several parameters come into play in terms of elucidating the awakening churches. Firstly, these churches promote the idea of autonomy from foreign religions, and they foster the sense of ownership of spiritual and social space. Secondly, the awakening churches offer a space for freedom of speech and maneuvers to transform one's life condition. Thirdly, they are vehicles of a new world order where everything is possible with Jesus. Fourthly, they are religious movements that influence the country's political life in many different ways, including political ideologies and electoral periods. Fifthly, the awakening churches reflect a missionary impetus. Lastly, they develop a new ethos of prosperity in which misery is perceived as the work of the devil in people's lives. (Fallut 2012, pp. 45–46).

From another perspective, a Congolese circle of reflection has documented sixteen main objectives of the awakening churches following its interviews with ten of them. I cite its report in full below. The awakening churches seek to carrying out the following actions:

1.    Evangelize and exercise the Pastoral Ministry under the inspiration of the Holy Spirit;
2.    Build the parishes;
3.    Create social works: (medical centers, Bible schools, kindergartens, primary schools, secondary schools, professional schools, universities, breeding, agriculture, etc.) for the well-being of the population;
4.    Prepare the world for the second coming of Christ by proclaiming the full gospel by all means;
5.    Promote the progress and expression of the charismatic movement;
6.    Develop communities through the creation of cooperatives and educational, social, health, agricultural, and philanthropic works;
7.    Preach the message of God;
8.    Teach the mysteries of the Bible;
9.    Celebrate services, administer baptism according to the Bible, consecrate marriage, and give Holy Communion;
10.   Create framework conditions for life aimed at improving the spiritual state of the human person to improve his behavior towards his neighbor and society in general, and God in particular;
11.   Challenge the conscience of man by reminding him that, at the same time, he must work for the development of his immediate environment, and he must prepare for his beyond since life does not stop with death;
12.   Inform public opinion and, above all, make it aware of the Promises of the Bible fulfilled in this time of the end;
13.   Teach the masses;
14.   Teach the Good News of Christ, as rendered by the Holy Scriptures;
15.   Contribute to the improvement of morals and the development of human society through moral and Christian precepts and virtues, as well as through social and charitable works, providing assistance to the deprived community (schools, hospitals, agricultural exploitation and breeding, popular education by radio and television);

16. Empower the whole person through community development. (La Bibliothèque Virtuelle de la République Démocratique du Congo 2015)

From my standpoint, in agreement with the above description, I think that one fact may be the overarching distinguishing feature of these Christian movements. Compared to Martin Luther's *sola scriptura*, the postulate of all the awakening churches is *solus Spiritus Sancti*. They claim that the unreserved reliance on the Holy Spirit dictates everything that they believe and carry out. In these circumstances, Shako Lokeso observes that "It is not surprising and even rare to hear, in Congo-Kinshasa, many of the pastors proclaiming that only inspiration is enough to understand the Bible. The Bible does not need to be studied." (Shako Lokeso 2018, p. 61). As a result, it is commonplace to hear the pastors of awakening churches insist on Jesus' lesson to Nicodemus: "The wind blows where it wills, and you can hear the sound it makes, but you do not know where it comes from or where it goes; so it is with everyone who is born of the Spirit."[9] They use this Gospel passage to ground their unmediated vocation from God, in the same way that the mystical experience was understood as "*cognitio Dei experimentalis*, that is, experimental knowledge of God." (See (Alois-Maria and Matthias 1998, pp. 265–66)).

Moreover, the *solus Spiritus Sancti* is also the basis of their considerable latitude of speech and deeds. On this account, to prevent people from questioning their actions, they use as a weapon another Gospel passage which says, "Every sin and blasphemy will be forgiven people, but the blasphemy against the Spirit will not be forgiven." (Matthew 12: 31). In other words, since the Holy Spirit works in and through them, nobody has a right to question or challenge their thinking, choices, and conducts.

It is under these circumstances that I find the angel of light at work in the approach of the awakening churches.[10] In the fourth rule (rule 332) of the Second Week of the *Spiritual Exercises*, Saint Ignatius of Loyola describes how the devil works as an angel of light to win over the soul of a Christian. He notes, "It is characteristic of the evil angel, who takes on the appearance of an angel of light, to enter by going along with the devout soul and then to come out by his own way with success for himself. That is, he brings good and holy thoughts attractive to such an upright person and then strives little by little to get his own way, by enticing the soul over his own hidden deceits and evil intentions." (Ignatius of Loyola 1991, p. 206).

In other words, Saint Ignatius draws attention to the fact that the devil does not tempt devout persons like pastors with graphic solicitations or enticements. He uses cunning stratagems to trap such persons. He used the same deceptive tactics to tempt Jesus but he failed.[11] Today, the devil is using the same temptations of materialism, power, and glory to allure pastors of the awakening churches, and sadly many are falling.[12] Cardinal Karol Wojtyla helps us to understand why those pastors are succumbing. He explains the devil's approach in this way:

"Satan does not achieve complete victory; he shows himself incapable of sowing the seed of total rebellion in man, that total rebellion of which he himself is the expression. Instead, he succeeds in inducing man to turn toward the world, and to stray progressively in a direction opposed to the one to which God has called him. From that moment the world becomes the terrain of man's temptation: a terrain where man turns his back on God; a terrain of rebellion rather than of collaboration with the Creator; a terrain where human pride seeks not the glory of God but its own greater satisfaction." (Wojtyla 2021, p. 34)

This state of affairs is verified when, for example, instead of having a Christocentric approach to their ministries of preaching, healing, and the well-being of people, many pastors of the awakening churches put Jesus aside and become self-centered. They, thus, preach themselves, and they go after wealth, power, and fame.[13] As an angel of light, the devil seduces pastors to lead them to miss the mark[14] of their vocation, which is, as expressed by Saint Ignatius in the Principle and Foundation of the *Spiritual Exercises*, "to praise, reverence, and serve God our Lord, and by means of this to save their souls."[15]

After discussing the nature of the *églises de réveil* and indicating how the angel of light is at work to deride this nature, the ensuing section details how the angel of light has asserted his will in these awakening churches.

## 3. The Modus Operandi of the Eglises de Réveil

Following the analysis of the nature of the awakening churches, this section gives a narrative that elucidates their *modus operandi*. This narrative begins by pointing out the ubiquity of the awakening churches. In the DRC, awakening churches fill almost every street, neighborhood, or borough of a city. It suffices to take a short walk in the late afternoon or walk around on Sunday morning to realize it. As Shako Lokeso reports

> "In the Democratic Republic of the Congo, there is a real enthusiasm for religion today. Alongside the traditional institutional churches, sects and different groups or religious movements are born and develop at a speed defying all statistics. If in the 1990s, these sects and religious groups were limited to the big towns and cities, today they abound throughout the national territory. They are scattered from the capital to the most remote and landlocked villages. The rate of their proliferation is such that no one, even at the level of Public Administration, can advance any figure." (Shako Lokeso 2018, pp. 55–56)

This impressive and dizzying proliferation of the *églises de réveil* does not reflect authentic Christian witness. The motives for this proliferation are for the most part non-evangelical. For instance, there are countless churches with various names, including "Come and See", "Prophetic Consultation", "Light of the World", "God of Sign", "Jericho Trumpet", "Missionary Agency of Deliverance", "The Place of Salvation", "Apostolic Church—Fire of Awakening", "The Church of God—The Awakening", "Evangelical Church of Awakening", "Evangelical Church—The New House of David", etc. The list of these churches can be endless, and their names already imply competition, if not heterogeneity of purposes.

I diagnose in this context a *modus operandi* that is commensurate with an African politician's lifestyle. From my perspective, three features characterize an African politician's life: worldliness, nepotism, and aggressiveness.[16] Firstly, worldliness encompasses the wealth, fame, and power that an African politician seeks to acquire by entering politics.[17] Secondly, according to the *New Oxford American Dictionary*, nepotism is "the practice among those with power or influence of favoring relatives or friends, especially by giving them jobs." (*New Oxford American Dictionary* 2005–2021). When in power, an African politician wants to be surrounded by family members and close friends. He, thus, hires these relatives to work as janitors or housekeepers, gardeners, cooks, barbers, drivers, sentinels, bodyguards, secretaries, etc.[18] Lastly, Max Weber pointed out that "The decisive means for politics is violence." (Weber 1946, p. 121). An African politician fiercely uses this means to eliminate his challengers and secure his interests.

Interestingly, this political approach has become the *modus operandi* of many pastors of the *églises de réveil* in both urban and rural areas. To account for this statement, the following narrative indicates how this *modus operandi* works in urban and rural areas.[19]

My experience of taking the bus from 2003 to 2005 and nowadays in Kinshasa (the capital city of the DRC) has taught me that many itinerant preachers experience unemployment and joblessness. Religion becomes a means for another end, namely subsistence. Under these circumstances, such itinerant preachers are not religious animals. According to W. E. B. Du Bois, a religious animal is "a being of that deep emotional nature which turns instinctively toward the supernatural." (Du Bois 2007, p. 135). In the context of itinerant preachers, religion is not constitutive of their being but instrumental in securing their survival (See (Simantoto Mafuta 2019, pp. 33–42)). That is why every preaching session on the bus ends with sugary stories of how God marvelously blesses people who support His servants. Afterward, the preacher holds a basket or opens his hands to ask for a generous collection.

In addition to this attitude of itinerant preachers, the management of the pastors of *églises de réveil* in urban areas arouses suspicion because they entertain confusion between a church's holdings and a family's businesses or foundations. These pastors write church estates under their names.[20] Moreover, while they postulate *solus Spiritus Sancti*, the succession of pastorship is often hereditary. People, thus, wonder if the charisma of being a pastor is transferable like genes. Another issue that raises concern in the *modus operandi* among the pastors of the awakening churches is the *maisons des dons* (houses of gifts). Rumor has it that devotees who want to be prosperous have to enter occult or esoteric *maisons des dons* owned by their powerful pastors. In return, they are expected to make costly sacrifices when they become affluent.

If money and power are the focus in urban areas, wealth and social status constitute the attention of rural pastors of awakening churches. To achieve this goal, these pastors are deeply versed in the practices of exorcism ministry. With dramaturgical and unusual practices, they act as thaumaturges to attract countless adepts because the social status and wealth of a pastor depend on the "enrollment" of people in a particular church. This state of affairs bewilders people. Consequently, many people from the villages currently ridicule the *églises de réveil* phenomenon by claiming that former witch doctors have become pastors because the business of the latter has no downtime and it pays better.[21]

Overall, aggressiveness appears to be an effective medium for evangelization in the *églises de réveil*.[22] Like a politician who does not spare his opponents during campaigns to secure votes and victory in the time of the election, pastors of awakening churches disparage with brutal virulence their competitors to attract a good number of followers. In the mindset of aggressiveness against any competitors, prayer gatherings of many *églises de réveil* involve commotion and cacophonic glitz. It is not uncommon that three people use big loudspeakers while they pray. Under these circumstances, anybody who raises concern about the disturbance of the peace at night is labeled by pastors of awakening churches as being under demonic influence, allergic to prayer, and hostile to the Gospel of Jesus Christ. This labeling and demonization are their mechanisms for silencing their rivals and critics.

Moreover, according to the philosophy of many pastors, having many followers mean a good collection, and a profuse collection represents a good life for them and their families. To increase the flow of funds, some pastors have creatively introduced social stratification during prayer gatherings. The members are encouraged to purchase seats in the church. As a result, the highest bidders occupy the most strategic and prestigious places in the church.[23]

In addition, the preaching of many pastors of the *églises de réveil* is empty or does not have much to offer in terms of Christian doctrine.[24] Here also, pastors are like politicians who, bereft of political will and vision, are incapable of rationally and coherently defending a societal project. Their preaching is monotonous.[25] The conclusion of this preaching is the same: the more money a person gives to the church, the greater will be her blessings in this life. In other words, the prosperity Gospel constitutes the sole content of this type of preaching. The Gospel of the Cross is absent. This preaching is unlike Saint Paul's proclamation of the Gospel to the Corinthians. He avows that "we proclaim Christ crucified, a stumbling block to Jews and foolishness to Gentiles, but to those who are called, Jews and Greeks alike, Christ the power of God and the wisdom of God." (1 Corinthians 1: 23–24). Later, he reminds the Corinthians, "When I came to you, brothers, proclaiming the mystery of God, I did not come with sublimity of words or of wisdom. For I resolved to know nothing while I was with you except Jesus Christ, and him crucified." (1 Corinthians 2: 1–2).

Besides the focus on multifaceted prosperity, healing and deliverance from curse, bondage, and bad luck are addressed in the preaching for the sake of drawing more followers who will give big donations in the long run. For example, Shako Lokeso indicates that "illness, sterility, death, lack of husband or wife, lack of material prosperity, [seeking] a visa to [go] abroad are considered consequences on the part of sorcerers. The pastors, themselves, contribute to valuing these beliefs and to make their followers live under the psychosis of the sorcerers who would attack their life." (Shako Lokeso 2018, p. 63). The

pastor promises to deliver them from misfortune provided that they show gratitude, that is, give him plentiful donations.

In a similar fashion, Soiron Fallut draws attention to one essential hallmark of the *églises de réveil*. She writes,

> "Beyond the possibility of a new form of sociability at the family level and in the social body as a whole, the resolution of material problems is often the first reason that pushes new believers into these awakening churches. Considered "everything of life" churches, these churches, through the pastors, were therefore first attended by the less privileged social classes, in a search for concrete solutions, in terms of health, work, or affective relationships. The quest for a cure is often the trigger. Based on a literal reading of the Bible (and the verse from the Gospels indicating: "In my name, they will cast out demons, they will lay on their hands and the sick will be healed"), pastors, thanks to their gifts are supposed to heal their devotees. These churches, therefore, have pragmatic aims. The temporal effectiveness of their relationship to the sacred is predominant."
> (Fallut 2012, pp. 43–44)

Put differently, these awakening churches would die out if social institutions like the State, the market, or civil society secured holistic health or well-being for the population.[26]

The foregoing narrative has shown how the angel of light has worked in the life of the awakening churches, especially their pastors. Their *modus operandi* is unsuccessful and counterproductive. Instead of living out a Christocentric life, many pastors of the awakening churches have succumbed to autolatry and idolatry of worldliness (wealth, power, and fame). The last section intends, therefore, to propose a perspective for authentic Christian growth today.

## 4. Perspective for Authentic Christian Growth

After refuting the *modus operandi* of the pastors of many *églises de réveil* in the previous narrative, the subsequent reflection proposes a perspective for authentic Christian growth. Today, any Christian mission needs to reflect radical Christocentrism. Beyond traditions and denominations, any Christian witness has to radically establish Jesus as the measure of everything, including thoughts, choices, and deeds. Christians are challenged to proclaim Jesus and nothing else or nothing more. As Wojtyla expresses it, "He [Jesus] is, in himself, the full and final measure of the mystery of [humankind], so profound and yet so simple. All of us are servants of that mystery, the mystery of Christ, a great prophet!" (Wojtyla 2021, p. 137). In other words, Jesus always remains a role model for today's Christians.

An illustration can make this statement intelligible and compelling. During his earthly life, Jesus' wholehearted focus was on the mission He received from His Heavenly Father. In the Gospel of Saint John, "sentness" permeates Jesus' whole life and ministry.[27] Brad Brisco argues that "The entire Gospel is about sending and being sent. The term 'sent' and its derivatives appear almost sixty times in the Gospel of John." (Brisco 2012). Jesus had a keen and vivid consciousness of his identity as the One sent by God. The scope of this reflection precludes giving an exhaustive account of this Johannine theme of Jesus' sentness.[28] Two Gospel passages suffice to make the point. These passages involve Jesus' vertical and horizontal relations.

The first passage relates to Jesus' vertical relation, that is, his mission before the Father. In John 17, which can be considered as the "stocktaking" of Jesus' mission, the verb *send* occurs seven times, namely in verses 3, 8, 18 (twice), 21, 23, and 25. These verses teach that Jesus always felt accountable to the One who sent Him.[29] His words and actions unambiguously reflect this sense of accountability.

The second passage concerns Jesus' commission to His Apostles. In John 20: 21, He says, "Peace be with you. As the Father has sent me, so I send you." Raymond Brown sees this verse as both the paradigm of the mission and a Johannine theological theme. He observed that "We may compare it to [John 17: 18]: 'As you sent me into the world,

so I sent them into the world." (Brown 1970, p. 1030). As Neal Flanagan interprets it, "Simultaneously, he sends out these disciples just as the Father had sent him (v. 21). His mission becomes theirs; his work is placed in their hands. And that mission, that work, is to manifest God who is love in their words and deeds. Through them now, enlivened by the Spirit, will the presence of God become known and seen and felt in the world. If in truth Jesus is God's sacrament, God's exegete, we in turn through the Spirit become Jesus' sacraments, his living exegetes." (Flanagan 1983, p. 96). Jesus was the missionary of his Heavenly Father. To Philip and other Apostles, he says, "Have I been with you for so long a time and you still do not know me, Philip? Whoever has seen me has seen the Father." (John 14: 9). In a similar manner, as missionaries of Jesus, pastors of the *églises de réveil* should help modern men and women to see and encounter Jesus as did the Apostles, particularly Philip and Andrew.[30]

The foregoing analysis has helped us to understand how Jesus remains a role model for today's Christians. His earthly life was a "Father-centric" life. Jesus never betrayed the Father who sent Him. In John 5: 30, He declares, "I cannot do anything on my own; I judge as I hear, and my judgment is just because I do not seek my own will but the will of the one who sent me." Similarly, in John 6: 38, he stresses, "I came down from heaven not to do my own will but the will of the one who sent me." This attitude of Jesus, therefore, challenges Christians to lead Christocentric lives. As mentioned above, Saint Paul led a Christocentric life. To the Corinthians, he declared, "I resolved to know nothing while I was with you except Jesus Christ, and him crucified." (1 Corinthians 2: 2). Later, he reveals to them the cost of his Christocentric discipleship in these terms:

> "Five times at the hands of the Jews I received forty lashes minus one. Three times I was beaten with rods, once I was stoned, three times I was shipwrecked, I passed a night and a day on the deep; on frequent journeys, in dangers from rivers, dangers from robbers, dangers from my own race, dangers from Gentiles, dangers in the city, dangers in the wilderness, dangers at sea, dangers among false brothers; in toil and hardship, through many sleepless nights, through hunger and thirst, through frequent fastings, through cold and exposure. And apart from these things, there is the daily pressure upon me of my anxiety for all the churches. (2 Corinthians 11: 24–28)."[31]

Despite these hardships, Saint Paul did not deviate or waiver from bearing witness to Jesus Christ. Like him, today's Christians, especially pastors of the *églises de réveil*, are called to be loyal to Jesus Christ and faithful to their Christian identity and mission. A sincere and more careful meditation on the Gospel can help them to acquire the *sensus Christi* that will eventually correct their current *modus operandi*.[32]

After explaining how Christians can learn from Jesus, my reflection singles out two levels of this radical Christocentrism, namely a sign of contradiction vis-à-vis the world and a radical witness of the Gospel of Jesus Christ. Without a doubt, being a sign of contradiction can constitute the appropriate way to be a radical witness to the Gospel, but my analysis suggests specific orientations for the two dimensions of a Christian life.

### 4.1. The Sign of Contradiction

The word "contradiction" is philosophically and theologically laden. Generally, as a noun, the word "contradiction" has three nuances. Firstly, contradiction means "a combination of statements, ideas, or features of a situation that are opposed to one another." Secondly, contradiction represents "a person, thing, or situation in which inconsistent elements are present." Thirdly, contradiction means "the statement of a position opposite to one already made."[33]

In philosophy, Patrick Grim notes that "The notion of contradiction is far from simple, it turns out, and the search for clarification points up a menagerie of different forms of the Law of Non-Contradiction and Dialetheism as well." (Grim 2004, p. 49). A thorough discussion of this complexity necessitates going far afield. Here, it suffices to note that,

according to Grim, "The Law of Non-Contradiction holds that both sides of a contradiction cannot be true. Dialetheism is the view that there are contradictions both sides of which are true." (Grim 2004, p. 49). Aristotle, the classic theoretician of the law or the principle of non-contradiction, believes that this principle is the most certain of all principles, and it is by nature the starting point of all the other axioms (Aristotle 1933, pp. 161, 163). He, thus, expresses this principle of non-contradiction as follows: "It is impossible at once to be and not to be." (Aristotle 1933, p. 163). His modern interpreter Paula Gottlieb points out that "There are arguably three versions of the principle of non-contradiction to be found in Aristotle: an ontological, a doxastic and a semantic version. The first version concerns things that exist in the world, the second is about what we can believe, and the third relates to assertion and truth." (Gottlieb 2019). According to her, Aristotle expresses these versions in the following way: "The first version [...] is usually taken to be the main version of the principle and it runs as follows: 'It is impossible for the same thing to belong and not to belong at the same time to the same thing and in the same respect.'" (Gottlieb 2019). The second version is as follows: "It is impossible to hold (suppose) the same thing to be and not to be.'" (Gottlieb 2019). The third version is that "opposite assertions cannot be true at the same time.'" (Gottlieb 2019).

In light of this philosophical notion of the principle of non-contradiction, a person cannot belong to Jesus and not belong to Him at the same time and in the same respect. They cannot be a Christian and a non-Christian. They cannot believe in the Gospel and reject it at the same time.

The theological perspective of contradiction relates to the prophetic words of the elderly Simeon to the Virgin Mary in the entryway of the Jerusalem Temple: "Behold, this child is destined for the fall and rise of many in Israel, and to be a sign that will be contradicted (and you yourself a sword will pierce) so that the thoughts of many hearts may be revealed."(Luke 2: 34–35). I am indebted to Cardinal Karol Wojtyla's meditation on this passage. He believes that "the words 'a sign of contradiction' sum up most felicitously the whole truth about Jesus Christ, his mission, and his Church." (Wojtyla 2021, p. 211). He admirably explains this statement in the following way:

> "Any objective examination of the Gospel shows Jesus Christ to have been above all a teacher of truth and a servant of love, and it is these characteristics of his which explain the real meaning of all that he did and all that he set out to do. They also explain both the contradiction inherent in his mission and activities and the contradiction aroused by the teaching and behavior of the teacher from Nazareth. Jesus disputed the totally mistaken and false interpretation of the word of God and the tradition of the chosen people that were upheld by the Pharisees and Sadducees. He opposed whatever was not in keeping with the primary and fundamental truth of the Word. He opposed all the petty human meanness that was distorting the Law and the greatest of all the commandments, the law of love. He opposed them not only in what he said but in what he did. His teaching consisted above all in the works he performed, "all that Jesus began to do and teach (Acts 1: 1). He never intended this opposition, this contradiction, to have any political implication. "Render to Caesar the things that are Caesar's and to God the things that are God's" (Mt 22: 21; Lk 20: 25)." (Wojtyla 2021, p. 96)

This commentary demonstrates how Jesus' life and ministry contradict anything contrary to the identity of God. The Word became flesh to reveal the true face and will of God. This attitude of Jesus is the comportment that today's Christians need to emulate. In other words, it means following the example of Jesus, to be a sign of contradiction does not mean to be a lawless person. It supposes rejecting anything incompatible with Jesus, including evil, sin, social ills, and idolatry. The hymn of peace of Saint Francis of Assisi encapsulates the quintessence of this idea of being a sign of contradiction today. It teaches that love contradicts hatred, forgiveness contradicts wrongness, harmony contradicts discord, truth contradicts error, faith contradicts doubt, hope contradicts despair, light

contradicts shadows (darkness), joy contradicts sadness, and life contradicts death (Prayer of Saint Francis n.d.).

From this perspective, to be true to their Christian vocation, all Christians, and in particular the *églises de réveil*, need to radiate love, forgiveness, harmony, truth, faith, hope, light, joy, and life. When they live out these Christian virtues, they refute hatred, wrongness, discord, error, doubt, despair, darkness, sadness, and death, which pollute and suffuse the world. My analysis, thus, concurs with Raymond Brown when he writes, "If the disciples are *sent* by Jesus into the world, it is for the same purpose for which Jesus was sent into the world—not to change the world but to challenge the world. In each generation, there is on earth a group of men given by God to Jesus, and the task of the disciples is to separate these sons of light from the sons of darkness who surround them. Those given to Jesus will recognize his voice in and through the mission of the disciples and will band together into one." (Brown 1970, p. 764).

The foregoing viewpoint can raise an issue related to the truth that "God so loved the world that he gave his only Son so that everyone who believes in him might not perish but might have eternal life." (John 3: 16). To clear up this possible misunderstanding, I point out that in the Johannine Gospel, the idea of the world is ambivalent. As H. Richard Niebuhr observes, "One of the apparent paradoxes of the Fourth Gospel is that the word 'world,' so used for the totality of creation and especially of humanity as the object of God's love, is also used to designate mankind in so far as it rejects Christ, lives in darkness, does evil works, is ignorant of the Father, rejoices over the death of the Son. The ruler of the world is not the Logos but the devil. Its principle is not truth but the lie; it is the realm of murder and of death, rather than of life." (Niebuhr 2001, p. 198). Put differently, on one hand, the world denotes the work of God's love. It is His creation. That is why God is willing to sacrifice his Only-Begotten Son to save it. (John 3: 16; 6: 51). On the other hand, the world represents the haven or den of the powers that oppose the eternal will and plan of God. As I interpret this Fourth Gospel, especially John 17, the world as the hostile force against God is larger than the world that is a part of creation.[34] The interpretation of H. Richard Niebuhr helps to capture the scope of the meaning of the world in this religious context. He writes,

> "The injunction to Christians is, "Do not love the world or the things in the world. If anyone loves the world, love of the Father is not in him." [1 John 2: 15]. That world appears as a realm under the power of evil; it is the region of darkness, into which the citizens of the kingdom of light must not enter; it is characterized by the prevalence in it of lies, hatred, and murder; it is the heir of Cain. It is a secular society, dominated by the "lust of the flesh, the lust of the eyes and the pride of life", or in Prof. Dodd's translation of these phrases, it is a "pagan society, with its sensuality, superficiality and pretentiousness, its materialism and its egoism." It is a culture that is concerned with temporal and passing values, whereas Christ has words of eternal life." (Niebuhr 2001, p. 48)

It is worth noting that Christians cannot relate to the world with fear and trembling like Saint Paul did vis-à-vis the Corinthians. (1 Corinthians 2: 3). Neither are they to be naïve. As it stands today, the world is not neutral.[35] That is why, in his priestly prayer, Jesus asks His Father to consecrate His disciples in Truth. As Pope Benedict XVI explains it, "Consecration means the total claim of man on the part of God, a 'separation' for Him, which, however, is at the same time a mission for the peoples." (Ratzinger 2014, p. 458). Being separated for God implies protection from the world's darkness and lies. Pope Benedict XVI writes, "Consecrate [sanctify] them in the truth. This means: immerse them completely in Jesus Christ so that what Paul indicated as the fundamental experience of his apostolate is verified for them: 'It is no longer I who live, but Christ who lives in me' (Ga 2, 20)." (Benoît and Sarah 2019, p. 70).

In light of the above analysis, to be a sign of contradiction today, Christians in general, and the members of the *églises de réveil* in particular, need to have authentic attitudes, such

as *Parrhesia* (boldness),[36] self-criticism, moral and spiritual discernment, and "positive suspicion" or lucidity, vis-à-vis the pressures and solicitations of today's world.

After delineating the scope of the sign of contradiction as the first level of the radical Christocentrism that I propose in this essay, the subsequent paragraphs consider its second level.

### 4.2. Radical Witness to the Gospel of Jesus Christ

In December 1975, Pope Paul VI drew attention to an unambiguous truth. He stated that "Modern man listens more willingly to witnesses than to teachers, and if he does listen to teachers, it is because they are witnesses." (Paul VI 1975, §41). What, therefore, is the Christian witness?

Negatively speaking, the witness is not identical to proselytism, which often involves "coercion of people's consciences." (Ratzinger and Habermas 2006, p. 43). Rather, witness consists of the readiness for giving an account of one's religious convictions with gentleness, reverence, and purity of conscience (1 Peter 3: 15–16). It means walking in the footsteps of Jesus who proclaims the truth and imminence of God's *basileia* (reign or kingdom). As Jacob Neusner observes, "If I had to point to one thing to which Jesus would surely point, it is the kingdom of heaven, which, he held, was soon to come into being: Over and over again, Jesus tried to explain what was at stake: 'Repent, for the kingdom of heaven is at hand' was Jesus' first message (Matt 4: 17). So at stake is overcoming sin, so as to enter God's kingdom." (Neusner 2000, pp. 120–21). This observation shows that the experience of the kingdom of God depends on conversion. From Jesus' perspective, sin constitutes the overwhelming obstruction or paralysis that prevents people from living in the presence of God and enjoying the fullness of His life. Pope John Paul II rightly describes the nature of sin as a suicidal act.[37] As a result, bearing witness to Jesus has to address the issue of sin in individuals' lives and society. When four men brought a paralytic to Jesus, He said to him, "Child, your sins are forgiven." (Mark 2: 5). Far from being a blaspheme, this utterance achieved a double objective. Firstly, it revealed the divinity of Jesus.[38] Secondly, it underlined the value scale in the realm of mission. As Pope Benedict XVI interprets it,

> "That Jesus is able to forgive sins, he shows it now by ordering the sick to take up his mat and go away, healed. But yet he thereby safeguards the priority of the forgiveness of sins as the foundation of all true human healing. Man is a relational being. If the first of these, man's fundamental relationship—the relationship to God—is disrupted, then nothing else can really be right. It is this priority that is in question in the message and the action of Jesus. He wants, first of all, to draw man's attention to the heart of his evil and show it to him: if you are not healed in this [matter], then, despite all the good things you may find, you will not really be cured. In this sense, the explanation of the name of Jesus given in a dream to Joseph already sheds fundamental light on the way of conceiving the salvation of man and shows what consists, therefore, the essential duty of the one who brings salvation." (Ratzinger 2014, pp. 46–47)

This commentary teaches that the mission of Christians, which is synonymous here with witness, is an *imitatio Christi*. Like their master and Lord, Christians are called to challenge themselves and their fellow modern people to renounce sin. After resetting the existential order through His paschal mystery, Jesus commissioned the Apostles with these words: "Receive the Holy Spirit. Whose sins you forgive are forgiven them, and whose sins you retain are retained." (John 20: 22–23). It is, therefore, a betrayal of their mission when, today, pastors of the *églises de réveil* de-emphasize the call to conversion and become obsessed with worldliness, that is, wealth, power, and fame. This obsession accounts for the banalization of sin and the minimization of the horror of evil in their society.

What is happening today is a scandal. The proliferation of social ills takes place at the same speed as the proliferation of the *églises de réveil*.[39] For example, it is unfortunate to note the reversal of influence between politics and religion. Whereas the Christian religion

influenced the politics and cultures that it encountered, politics manipulates and influences religion in places like the DRC today.[40] If a kinsman is a pastor of an awakening church, the members of his ethnic group are expected to support him by belonging to that church. That pastor will, in turn, mobilize them to support the ideology of their relative who is a politician. Sometimes, the pastor is at the same time a politician. In this case, his relatives are expected to aid and abet both his religious and political enterprise. This state of affairs is in total contrast to the witness of Christians of the early Church. As Martin Luther King reports,

> "In those days the church was not merely a thermometer that recorded the ideas and principles of popular opinion; it was a thermostat that transformed the mores of society. Whenever the early Christians entered a town, the people in power became disturbed and immediately sought to convict the Christians for being "disturbers of the peace" and "outside agitators." But the Christians pressed on, in the conviction that they were "a colony of heaven", called to obey God rather than man. Small in number, they were big in commitment. They were too God-intoxicated to be "astronomically intimidated." [Through] their effort and example, they brought an end to such ancient evils as infanticide and gladiatorial contests." (King 1963)

This account shows that Christian witness must be a ferment that challenges and transforms the status quo. Walter Brueggemann understands this witness in terms of an alternative community. He states, "In thinking this way, the keyword is *alternative*, and every prophetic minister and prophetic community must engage in a struggle with that notion." (Brueggemann 2001, p. 4). He adds, "So my programmatic urging is that every act of a minister who would be prophetic is part of a way of evoking, forming, and reforming an alternative community. And this applies to every facet and every practice of ministry." (Brueggemann 2001, p. 4) From this perspective, instead of imitating the systematic search for wealth, power, and fame that characterizes the *modus operandi* of politicians, the pastors of the *églises de réveil* need to form alternative communities where the hungry, the thirsty, the stranger, the naked, and the marginalized are welcomed and find the satisfaction of their existential needs.[41]

From this analysis, it emerges that Christian witness cannot be superficial. Jesus cannot be a means to accumulate wealth, power, and fame. He is the Omega, the end of Christian churches' ministries. Their sole goal is to lead people to Him to experience forgiveness, liberation, and salvation.

Furthermore, another way of understanding radical witness involves the perennial realities. Saint Paul teaches that "faith, hope, love remain, these three; but the greatest of these is love."[42] Simply put, the culture of faith, hope, and love encapsulates radical witness. Now, an in-depth description of each of these virtues will carry far afield the present analysis. A few illustrations suffice to show how faith, hope, and love express radical witness.[43]

As a characteristic disposition of a Christian, faith in Jesus must always be the motivation for their action. The attitude of the Apostles Peter and John toward a man disabled from birth is a prototype of Christocentric faith. As the Acts of the Apostles report, "Peter said, 'I have neither silver nor gold, but what I do have I give you: in the name of Jesus Christ the Nazorean, [rise and] walk.'" (Acts 3: 6). This testimony is contrary to the practices of the aforementioned *maisons des dons* (houses of gifts) found in some *églises de réveil*. Faith in God rebuts the worship of idols. It involves unreserved obedience to the first commandment: "You shall not have other gods beside me. You shall not make for yourself an idol or a likeness of anything in the heavens above or on the earth below or in the waters beneath the earth; you shall not bow down before them or serve them." (Exodus 20: 3–5). Jesus expresses this sacred rule in terms of wholehearted service to one master: "No one can serve two masters. He will either hate one and love the other, or be devoted to one and despise the other. You cannot serve God and mammon." (Matthew 6: 24). H.

Richard Niebuhr keenly understands the implications of the uncompromising loyalty to the true master when he writes,

> "Every Christian must often feel himself claimed by the Lord to reject the world and its kingdoms with their pluralism and temporalism, their makeshift compromises of many interests, their hypnotic obsession with the love of life, and the fear of death. The movement of withdrawal and renunciation is a necessary element in every Christian life, even though it is followed by an equally necessary movement of responsible engagement in cultural tasks. Where this is lacking, Christian faith quickly degenerates into a utilitarian device for the attainment of personal prosperity or public peace; and some imagined idol called by his name takes the place of Jesus Christ the Lord. What is necessary for the individual life is required also in the existence of the church." (Niebuhr 2001, p. 68)

The previous analysis sufficiently showed how many pastors of the *églises de réveil* have made religion "a utilitarian device for the attainment of personal prosperity." Unlike this attitude, when faith regulates thoughts and deeds, it requires both persons and churches to make choices that reject ungodliness and embrace what gives honor to God. In other words, bearing witness through faith means to have a "unique devotion to God."[44]

Apropos of hope, Jesus is again the paragon. Niebuhr observes that "Jesus' hope was in God and for God." (Niebuhr 2001, p. 21) Jesus' teaching on the dependence on God (Matthew 6: 25–34) is illustrative of His witness to hope. His attitude, thus, challenges current pastors who have attached their hope to wealth, fame, and power. To live a life full of hope means being nourished by an eschatological consciousness. As Pope John Paul II points out,

> "This truth which the Gospel teaches about God requires a certain *change in focus with regard to eschatology*. First of all, eschatology is not what will take place in the future, something happening only after earthly life is finished. *Eschatology has already begun with the coming of Christ*. The ultimate eschatological event was His redemptive Death and His Resurrection. This is the beginning of "a new heaven and a new earth" (cf. Rev 21: 1). For everyone, life beyond death is connected with the affirmation: "I believe in the resurrection of the body", and then: "I believe in the forgiveness of sins and in life everlasting." This is *Christocentric eschatology*." (John Paul II 1994, pp. 184–85)

From the perspective of hope, the best story that Christians across the board can tell through their speeches, choices, and deeds is that this world is inherently ephemeral and "creation awaits with eager expectation the revelation of the children of God." (Romans 8: 19). Under these circumstances, frugality, sobriety, moderation, self-control, and abnegation represent some virtues that convey a life full of hope. This approach discourages pastors to enslave their followers for their selfish prosperity. As Thomas Kelly puts it, "It is no longer for the glory of the church that one works. Rather, the people of God work for the kingdom of God, as the Lord's Prayer has always reminded us. 'Thy kingdom come, thy will be done, on earth.'" (Kelly 2013, p. 60) Only in this way can the eschatological consciousness become the *telos* of Christian witness.

Lastly, concerning witness through love, foremost, it is worth acknowledging that love is the most overused word.[45] However, love is not a vague or ambiguous reality as it is portrayed today. Nor is love a mere romantic sentiment. It is rather a decision and choice for God's glory and the well-being of others. That is why I find compelling the accounts of Saint Paul, Saint Augustin, and Martin Luther King. Firstly, Saint Paul's hymn of love accurately delineates the scope and demands of love (1 Corinthians 13: 1–13). Next, Saint Augustine distinguishes two types of love that build two different cities. He states, "Accordingly, two cities have been formed by two loves: the earthly by the love of self, even to the contempt of God; the heavenly by the love of God, even to the contempt of self. The former, in a word, glories in itself, the latter in the Lord. For the one seeks glory from men; but the greatest glory of the other is God, the witness of conscience. The one lifts its head

in its own glory; the other says to its God, 'Thou art my glory, and the lifter up of mine head.'" (Augustine of Hippo n.d.) Saint Augustine's observation is a powerful criterium of discernment when bearing witness to love. A Christian pastor who preaches the Prosperity Gospel, because it benefits him, builds his earthy life to the contempt of God. However, the one who denies himself, carries his cross, and follows the way of Jesus[46] loves God, and he glories himself in the Lord.

In addition, bearing witness to love involves telling people the truth about the necessity of conversion and holiness because "sin is the greatest evil since it strikes man in the heart of his personality. The first liberation, to which all others must make reference, is that from sin." (Congregation for the Doctrine of Faith 1984). From this perspective, Christian pastors no longer love their flocks when they preach politically correct homilies. Following Jesus' example, Saint Augustine teaches to condemn the sin but not the sinner.[47] Today, this fact is perversely misinterpreted in many circles of Christians. Indeed, Saint Augustine believes that "condemning sin but not the sinner" does not mean a *laissez-faire* attitude because "if He [Jesus] were a patron of sin, He would say, Neither will I condemn you; go, live as you will: be secure in my deliverance; how much soever you will sin, I will deliver you from all punishment even of hell, and from the tormentors of the infernal world. He said not this."[48] In the same account, Saint Augustine warns of two dangers of eternal perdition, namely the presumption of God's mercy and despair about God's forgiveness.[49] Such teachings, which have words of eternal truth, are uncommon in the preaching of many *églises de réveil*. This preaching loophole is not a service of love because people are deprived of teachings that translate Jesus' words of eternal life (John 6: 68–69; 11: 27). Whenever he preaches, a Christian pastor needs to put on the spirit of the martyr analogous to that of the ninety-year-old Eleazar and a mother and her seven sons recounted in the book of Maccabees.[50] Put another way, the spirit of the martyr "exalts the inviolable holiness of God's law"[51] in life's choices.

This idea of martyrdom is commensurate with Martin Luther King's idea of the creative extremism of love. He believes that Jesus was an extremist of love, and He expressed it in this way: "Love your enemies, bless them that curse you, do good to them that hate you, and pray for them which despitefully use you, and persecute you." King then elaborates as follows: "In that dramatic scene on Calvary's hill, three men were crucified. We must never forget that all three were crucified for the same crime—the crime of extremism. Two were extremists for immorality and thus fell below their environment. The other, Jesus Christ, was an extremist for love, truth, and goodness, and thereby rose above his environment. Perhaps the South, the nation, and the world are in dire need of creative extremists." (King 1963) To contextualize King's timely and burning proposal, I affirm that Christians in general, and the members of the *églises de réveil* in particular, need to become creative extremists of love in the way that I described above.[52]

All things considered, the discussion of this last section of my paper has considered a sign of contradiction vis-à-vis the world and absolute witness of the Gospel of Jesus Christ as a constitutive aspect of the radical Christocentrism that I proposed as an authentic way of being a Christian today.

## 5. Conclusions

This essay has qualified the affirmation that Africa is the hope of Christianity. While this fact might be true statistically, the quality of Christian living in this continent does not corroborate this assumption. My analysis pointed out that the *modus operandi* of many pastors of the *églises de réveil* (awakening churches) in Africa, especially in the DRC, supports my reservations. Being commensurate with an African politician's approach, this *modus operandi* revolves around worldliness, nepotism, and aggressiveness.

My analysis drew considerable insight from the Ignatian image of "the angel of light" to explain this sideslip. The father of lies (John 8: 44) tempts many pastors of the *églises de réveil* to control them and make their discipleship and mission fail. With a keen mind, Cardinal Karol Wojtyla perceives and indicates this malice of the devil when he states,

"The only thing that matters to him is to transmit to man his own rebellion, that is to say, the attitude with which he—Satan—has identified himself and by which he has, in consequence, placed himself outside the truth, which means outside the law of dependence on the Creator. That is the message of his '*non servam*' (I will not serve) (Jer 2: 20)." (Wojtyla 2021, p. 33) As mentioned above, many pastors' of the *églises de réveil* obsession with wealth, power, and fame is a potent expression of this "*non servam*" of the angel of light.

By itself, this perception marks a watershed in the process of establishing Africa as the hope of Christianity in the years to come. Nonetheless, I have proposed radical Christocentrism as an adequate perspective for the qualitative growth of the Christian faith in Africa, especially in the DRC. This radical Christocentrism must be lived through the attitudes of sign of the contradiction vis-à-vis the world and those of radical witness to the Gospel of Jesus Christ.

**Funding:** This research received no external funding.

**Conflicts of Interest:** The author declares no conflict of interest.

## Notes

1. (Statistica 2023). On behalf of the Catholic Church, J. J. Carney points out, "Catholics are everywhere. Nearly one out of two Christians—1.2 billion people in total—is a Roman Catholic. Of the ten most populous Christian countries in the world, four—the Philippines, Mexico, Brazil, and the Democratic Republic of the Congo—have majority Catholic populations. In every continent in the world, Catholics make up the largest single Christian family." (Carney 2022, p. 26). This prevalence of the Catholic Church makes intelligible the observation of Laurent Larcher, who says that "Among all the reasons for the [Catholic] Church's involvement in the crisis in the DRC [Democratic Republic of the Congo], we can put forward a final, more prosaic aspect. Rome must not 'lose' Africa, and above all, its youth. As we have seen, Africa is the continent where the number of Catholics increases the most each year, but it is also a continent where religious competition is intense. The Catholic Church faces serious competition from the Evangelical and Pentecostal churches (Protestantism is the main religious current on the continent) and Islam (Sunnis and Shiites) surpasses it. Africa is a continent where it [Catholic Church] must demonstrate its determination and its effectiveness in defending social justice and good governance, at a time when the new youth movements believe that it has failed in this area; when they do not consider her too timorous." See (Larcher 2018).

2. Readers are advised that I carried out the translation of all the French resources. In addition, I used the New American Bible, Revised Edition (NABRE), for Scripture quotes. Like Rabbi Jacob Neusner, I take seriously the Bible as the Word of God. (Neusner, pp. 10–11). Accordingly, I refer to biblical passages liberally and without much concern for the incommensurable opinions of biblical scholars on them.

3. In this essay, the adjective means *bona fide* or genuine. Jesus' compliment to Nathanael (John 1: 47) gives insight into my understanding of Christian authenticity. Authentic Christian witness is a common theme in African theological commentary on Christian growth on the continent. I think, for example, of the following works: (Ka Mana 2004) and (Katongole 2023). My approach may differ from the standard consensus because I emphasize a radical Christocentric life. A "Christocentric life" means putting Jesus at the center of a Christian's thoughts and deeds. It means to radically observe the greatest commandment that requires Christians to "love God with all your heart, with all your soul, with all your mind, and with all your strength." (Mark 12: 30). To use Saint Paul's words, a "Christocentric life" means "doing everything, in word or in deed, in the name of the Lord Jesus, giving thanks to God the Father through him." (Colossians 3: 17).

4. See (Okitafumba Lokola 2020; Lokola forthcoming). Other scholars have also studied the Catholic tradition in the Democratic Republic of the Congo. See, for example, (Carney 2010; Carney 2014a, 2014b; Kiess 2014; Katongole 2017; Onyumbe Wenyi 2020; Congo Research Group and Ebuteli 2022). Roger Alfani and Gérard Prunier have written valuable works on the Catholic tradition in the DRC from a political science perspective. See, for example (Prunier 2001; Alfani 2019).

5. Perhaps due to the language barrier, the experience of Pentecostalism in Francophone Africa is almost unknown in the Anglophone world. See, for example (Gifford 2004; Ogbu Kalu 2008; Marshall 2009; Asamoah-Gyadu 2013, 2015, 2020; Gifford 2016).

6. According to Pope Benedict XVI, "freedom is ever new. It is a challenge held out to each generation, and it must constantly be won over for the cause of good (cf. *Spe Salvi*, 24)." (Pope Benedict XVI 2008) This challenge has propelled me to rediscover and deepen the long-established truth of radical Christocentrism in missiology. Readers who are interested in knowing more about relevant contributions on this topic can peruse the following works: (Pope John Paul II 1190; Kraemer 2002; Muller 2006; Oumarou 2020; Verster 2021).

7. Mélanie Soiron Fallut concurs with my statement when she affirms, "If the number of churches and members is so difficult to establish, this is due in particular to the fact that a man who calls himself a pastor (without having always been ordained) retains his faithful only thanks to his charisma, his eloquence, and to his leadership abilities. So that the believer, if he is no longer

satisfied with the sermons, the prophecies, or the capacities of his pastor to manifest the divine power effectively, can decide at any time to change chapel, or even to create his own, without denying his beliefs." (Fallut 2012).

8    (Shako Lokeso 2018, p. 59) It is worth noting that in his analysis, Shako Lokeso puts in the same box sects and *églises de réveil*. In addition, readers can find powerful insights into the nature and scope of the awakening churches by perusing (Mukengeshayi Tshiaba Mbumba 2017).

9    John 3: 8. For the interpretation of this verse, readers can read (Doignon 1978, pp. 345–59).

10    It is worth emphasizing that I do not use the angel of light image to suggest that the awakening churches are satanic. Such a conception is utterly misleading. Saint Ignatius used it to help Catholic retreatants discern their inner motions toward or away from God. Hence, it is the case even today. I believe that this image is a way of drawing attention to the manner in which the devil tries to grip Jesus' disciples. In the Gospel, for example, Jesus said to Simon Peter, "Simon, Simon, behold Satan has demanded to sift all of you like wheat, but I have prayed that your own faith may not fail; and once you have turned back, you must strengthen your brothers." (Luke 22: 31–32). This account shows that temptations may occur even without a person's choice. That is why Jesus' disciples must always be vigilant and lucid. In addition, my argument in this article does not imply exclusivist ecclesiocentrism that is "the fruit of a specific theological system or a mistaken understanding of the phrase *extra ecclesiam nulla salus*." I share the conviction that "Exclusivist ecclesiocentrism is no longer defended by Catholic theologians after the clear statements of [Pope] Pius XII and Vatican Council II on the possibility of salvation for those who do not belong visibly to the [Catholic] Church (cf, e.g., Lumen Gentium 16; Gaudium et Spes 22)." See (International Theological Commission 1997, §10).

11    See the temptations of Jesus in Matthew 4: 1–11.

12    In the Gospel, Jesus adamantly affirms that a person cannot serve two masters (Matthew 6: 24–26). From this perspective, awakening pastors who establish businesses have fallen into the temptations of materialism, power, and glory. For example, Pastor Marcello Tunasi of *L'Eglise la Compassion* combines managing the Compassion Foundation with the MT vestments brand. Similarly, Pastor Moïse Mbiye of *Eglise Cité Bethel* combines managing the Moïse Mbiye Ministries with a music career and commercial activities. In addition, in *Combat Spirituel* of the late Maman Olangi, "the separation between what is called a Church and what is a family business is almost non-existent." See (Simantoto Mafuta 2019).

13    Some pastors name their churches after themselves. It is true in the case of the most impressive building of an awakening church in Kinshasa: Baruti Tabernacle. Others privatize God in the following way: God of [pastor] Lobandji, God of [Pastor] Sikatenda, and God of Neema. The name of one pastor is also revealing: General Sony Kafuta Rockman. Saint Paul already rebuked this state of affairs because it begets enmity and hostility. He disapproved of the jealousy and rivalry among the Corinthians when they claimed belonging to either him, Apollos, or Cephas. He saw their attitude as purely human. (1 Corinthians 3: 3–4). As a result, he urges them with these words: "So let no one boast about human beings, for everything belongs to you, Paul or Apollos or Cephas, or the world or life or death, or the present or the future: all belong to you, and you to Christ, and Christ to God." (1 Corinthians 3: 21–23). Furthermore, a follower of the Arc-en-ciel Church shared with me that his pastor Israel Panganga asserts that he cannot fast because he does not need God. Rather, God needs him. He brags that he did not look for God. It is God who sought after him. He also blesses the divorce when he says, "May God who blessed your union, now your separation." Likewise, Pastor Chiruza Zagabe of Eglise Primitive du Seigneur advocates polygamy among his followers. When they promote these social ills that contradict the Gospel teaching about the sacredness of marriage (Matthew 19: 4–6), pastors draw public attention to themselves. They, thus, preach themselves and put Jesus aside.

14    Missing the mark here is synonymous with a sin (transgression, trespass, offense, iniquity), understood in its Greek etymology as ἁμαρτία (*hamartia*), itself drawn from ἁμαρτάνω (*hamartánō*), which means "properly, to miss the mark (and so not share in the prize), and (figuratively) to err, especially (morally) to sin." See Blue Little Bible. Available online: https://www.blueletterbible.org/lexicon/g264/kjv/tr/0%E2%80%931/ (accessed on 11 July 2023).

15    (Ignatius of Loyola 1991, p. 130). Here, Ignatius emphasizes that "The other things on the face of the earth are created for the human beings, to help them in working toward the end for which they are created." It is worth noting that readers who have not witnessed what is going on in different *églises de réveil* in the DRC may consider my account unfair. As the fourth Gospel says, "An eyewitness has testified, and his testimony is true; he knows that he is speaking the truth, so that you also may [come to] believe." (John 19: 35). In addition to my testimony, I recommend reading (Mukengeshayi Tshiaba Mbumba 2017). This book title is revealing. It implies that, in the DRC, Jesus has become a business that swings between three options. These options are a revival in people's faith, their inducement to rapid sleep, and a shop for their pastors. Moreover, my analysis does not ignore the whole breadth of the Charismatic tradition, which has also influenced African Catholicism in great depth. In addition, the present analysis focuses on the *modus operandi* of the *églises de réveil*. That is why I do not, for example, allude to ways in which the Catholic priests have succumbed to such temptations. To illustrate, during his prayer meeting with Congolese priests, deacons, consecrated persons, and seminarians in early February 2023, Pope Francis told them that discipleship involves facing challenges and overcoming temptations. He pointed out three challenges or temptations prevalent in their specific context: spiritual mediocrity, worldly comfort, and superficiality. See Pope Francis, Address during prayer meeting with Congolese priests, deacons, consecrated persons and seminarians (Cathedral 2023). Accordingly, the danger of worldliness is not the exclusive problem of Charismatic pastors. Still, I argue that the lack of oversight and Prosperity Gospel worsen the temptation in the context of *églises de réveil*.

16 I do not fall in an overly simplistic generalization. I am aware that the political domain is very complex. However, my research and experience teach me that these three features characterize political life in Africa.

17 For more details on this subject, see (Bayart 2009).

18 It is intriguing to note that even one of the most ruthless dictators that the world has known, the former Congolese president Mobutu Sese Seko, denounced nepotism. See (Seko n.d.).

19 This narrative is construed from my personal experience of living in both urban and rural areas. Also, I attentively listen to people's appraisals of the different pastors. In addition, I listen to and watch their preaching.

20 It is beyond the scope of this paper to provide a catalogue of pastors of awakening churches who have written church estates under their names. Two emblematic examples are Olangi Wosho *Combat Spirituel* and Baruti Kasongo's *Tabernacle Baruti*. When the couple who founded *Combat Spirituel* died, there were discussions concerning the separation between business and church. The deceased's family was supposed to inherit the foundation, whereas the church community had to find a spiritual heir among the faithful. In addition, the issue of naming the church estate under the name of the chief pastor was among the main reasons for the schism between Pastor Baruti Kasongo and his associate Pastor Kalumbu Kiseka of *Assemblée Chrétienne de Righini*. Pastor Kalumbu Kiseka dissented when he found out that their church foundation became Baruti Kasongo's foundation and their church was named *Baruti Tabernacle*. Many other pastors have their own foundations, including Pastor Pascal Mukuna of *Assemblée Chrétienne de Kinshasa*, Pastor Léopold Mutombo of *Communauté Evangélique du Ministère Amen*, Pastor François Mutombo of *Communauté des Assemblées Chrétiennes voici l'homme* (CACVH), Pastor Moise Mbiye of *Eglise Cité Bethel*, Pastor Marcello Tunasi of *Eglise La Compassion*, and Pastor Joseph Omenga of *Place du Salut.*

21 I am indebted to my first-year students at Université de Lodja in the Democratic Republic of the Congo, who shared this repositioning during the class on *Sociologie Générale* during the Spring 2023 semester. During my lecture on the sociology of Karl Marx, I pointed out Marx's critique of religion as one source of humankind's alienation. When I asked them to evaluate Marx's claim that religion was the *opium of the people*, students made comments in all directions, in favor of the religion or against the blunders perpetrated by it. Some of them loosened their tongues to share the above-mentioned general impression.

22 In addition to what I mentioned in the main discussion, Apollinaire-Sam Simantoto Mafuta observes that "the slide into activism where verbal or physical violence is only one of the means of expression: hence his aggressive criticism towards non-members of the Church which is a response that the fanatic addresses to them." (Simantoto Mafuta 2019, p. 41). This characteristic attitude of the believer in *Combat Spirituel* is the pervasive medium for evangelization in the awakening churches. One pastor can never recognize the merit of another preacher because his followers would leave him and follow the one that he praises. See (Ingrid Kabongo TV n.d.)

23 I am grateful to my first-year students at Université de Lodja in the Democratic Republic of the Congo, who shared with me this practice during the class on Sociologie Générale during the Spring 2023 semester.

24 For example, Apollinaire-Sam Simantoto Mafuta illustrates this claim by pointing out the content of the doctrine of one awakening church. As he observes, "The doctrine of Spiritual Combat makes the Devil responsible for all the ills from which the Congolese suffer, notably poverty, unemployment, corruption, embezzlement, illness, celibacy, sterility, bewitchment, bad luck, etc." See (Simantoto Mafuta 2019, p. 35). This doctrine is negative, reductive, and depressing.

25 I argue that their followers do not perceive this monotony because, as Apollinaire-Sam Simantoto Mafuta puts it, this preaching has a numbing effect on them, and they are mentally manipulated and lulled. See (Simantoto Mafuta 2019, p. 38).

26 Alan WOLFE recognizes the existence of three major institutions in modern human society: the state, the market, and civil society. See (Wolfe 1989, p. 7).

27 I am indebted to Kim Hammond and Darren Cronshaw for this concept of sentness. See (Hammond and Cronshaw 2014).

28 To delve deeply into this theme, read the following resources: (Brown 1970, vol. 2; Stott 1975; Dunn 1981; Grob 1995; Waldstein 1990; Joosten 1997; Kostenberger and O'Brien 2001; Keener 2009).

29 John 6: 14–15 reports, "When the people saw the sign he had done, they said, 'This is truly the Prophet, the one who is to come into the world.' Since Jesus knew that they were going to come and carry him off to make him king, he withdrew again to the mountain alone." This passage admirably captures how Jesus never departed from the will of the One who sent Him.

30 As John 12: 20 reports, "Now there were some Greeks among those who had come up to worship at the feast. They came to Philip, who was from Bethsaida in Galilee, and asked him, 'Sir, we would like to see Jesus.' Philip went and told Andrew; then Andrew and Philip went and told Jesus. Jesus answered them, 'The hour has come for the Son of Man to be glorified.'".

31 For further insights into the cost of discipleship, see (Bonhoeffer 2018).

32 *Sensus Christi* means disposition understood as a "person's inherent qualities of mind and character." [*New Oxford American Dictionary*]. It emerges from Saint Paul's exhortation to the Philippians. He encouraged them in these terms: "Have among yourselves the same attitude that is also yours in Christ Jesus" (Philippians 2: 5). The former Superior General of the Society of Jesus admirably meditated on the scope of the *sensus Christi*. He called it "our way of proceeding." See (Arrupe 2004).

33 *New Oxford American Dictionary*.

34 For the sake of further persuasion, read (Niebuhr 2001, pp. 198–99).

35  On this account, I disagree with people who have advocated for the secularization of Christian values because the worldview of secularism and that of Jesus are not homogeneous. For example, the translation of religious (Christian) concepts into secular words remains an incalculable loss. Jürgen Habermas observes that "One such translation that salvages the substance of a term is the translation of the concept of 'man in the image of God' into that of identical dignity of all men that deserves unconditional respect. This goes beyond the borders of one particular religious fellowship and makes the substance of biblical concepts accessible to a general public that also includes those who have other faith and those who have none." (Ratzinger and Habermas 2006, p. 45). Notwithstanding the good intention behind the need for this translation, when human dignity loses its transcendent anchor, which is God, any human reasons fall short in its defense. My skepticism intensifies when Habermas affirms that "Indeed, a liberal political culture can expect that the secularized citizens play their part in the endeavors to translate relevant contributions from religious language into a language that is accessible to the public as a whole." (Ratzinger and Habermas 2006, pp. 51–52). This translation is a process of emptying the religious language of its deep spiritual meaning. Ben Zion Bosker is, thus, right when he asserts, "The secularization of culture has lowered human morale and wrought havoc with morality. Secular culture is anthropocentric in the grossest sense. By eliminating God as the center of existence, it leaves man alone to dominate the scene, with his empirical plans and purposes as the sole arbiter of life. He becomes the measure of all things and relates all values to his individual needs." (Bosker 1949, pp. 285–286). Today, it is a truism that secularism is an ideology of *pensée unique*. In this context, the claim of accommodation is an astuteness used to engulf the truth of the Gospel message. Accordingly, an accommodation to the secularism of this world is a wrong approach to the Christian mission. Of course, Christians are called to make intelligible their faith to today's people. In addition, I want to emphasize that I am aware of the nuances in the experience of secularism. However, the overarching characteristic of secularism across the board is the omission of God from human affairs. See (Taylor 2018).

36  A good example of this *Parrhesia* is the attitude of the man born blind when he lectured the Pharisees. See John 9: 10–34. According to the *Catechism of the Catholic Church*, *Parrhesia* involves "straightforward simplicity, filial trust, joyous assurance, humble boldness, the certainty of being loved." [The *Catechism of the Catholic Church*, §2778].

37  As he expresses it, "As a rupture with God, sin is an act of disobedience by a creature who rejects, at least implicitly, the very one from whom he came and who sustains him in life. It is therefore a suicidal act. Since by sinning man refuses to submit to God, his internal balance is also destroyed and it is precisely within himself that contradictions and conflicts arise. Wounded in this way, man almost inevitably causes damage to the fabric of his relationship with others and with the created world." (John Paul II 1984, §15).

38  According to the Archangel Gabriel, Jesus states that God saves His people from their sins (Matthew 1: 21; Luke 1: 31). As stated by Pope Benedict XVI, "The name of Jesus (*Jeshua*) means: YHWH is salvation. The messenger of God, who speaks to Joseph in a dream, explains what this salvation consists of: 'it is he who will save his people from their sins.' In this way, on the one hand, it is a very high theological duty entrusted to him, because only God himself can forgive sins. The child is thus placed in immediate relationship with God." (Ratzinger 2014, p. 45).

39  Several Congolese theatres have satirically denounced the many scandals of the awakening churches. For example, readers can find ample evidence by watching on YouTube all the episodes of *Ba Pasteurs Ya Kin* (The Pastors of Kinshasa), *Mondimi* (The Believer), and *Maman Bishop* by *Les Princes du Rire*, who originate from the Democratic Republic of the Congo.

40  See for example the analysis of (Onyumbe Wenyi 2020).

41  These categories of people constitute the criteria of the last judgment according to Saint Matthew 25: 31–46.

42  1 Corinthians 13: 13.

43  Readers can find more details about faith, hope, and love in the following documents: (Benedict XVI 2005, 2007; Francis 2013).

44  I borrowed Niebuhr's words. See (Niebuhr 2001, p. 27).

45  Pope Benedict XVI admirably discusses the polyvalence or polysemy of love at the beginning of his encyclical letter on Christian love. See (Benedict XVI 2005)

46  In Luke 14: 27, Jesus spells out the conditions of discipleship in these terms: "Whoever does not carry his own cross and come after me cannot be my disciple.".

47  Saint Augustine. Tractate 33 (John 7: 40–8: 11), 6. Available online: https://www.newadvent.org/fathers/1701033.htm (accessed on 31 July 2023).

48  Saint Augustine, Tractate 33 (John 7: 40–8: 11), 6.

49  Saint Augustine, Tractate 33 (John 7: 40–8: 11), 8.

50  See 2 Maccabees 6: 18–31; 7: 1–42. Pope Benedict XVI portrays a martyr in the following terms: "The Martyr is an exceedingly free person, free as regards power, as regards the world; a free person who in a single, definitive act gives God his whole life, and in a supreme act of faith, hope, and charity, abandons himself into the hands of his Creator and Redeemer; he gives up his life to be associated totally with the Sacrifice of Christ on the Cross. In a word, martyrdom is a great act of love in response to God's immense love." (Benedict XVI 2010).

51  I borrowed the words of Pope John Paul II. See (John Paul II 1993).

52  The analysis of this paper may disappoint readers who expected that I would give examples of some awakening churches that might be closer to the truth or those that can be "canonized." Further research will consider case studies of specific churches to meet such expectations. Meanwhile, I recommend reading (Katongole 2017). I reiterate that the objective of the present analysis was to qualitatively assess the overall *modus operandi* of the awakening churches, mainly focusing on the DRC.

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
