# Peer review of "The “Angel of Light at Work”: An Assessment of the Christian Mission in the Southern Hemisphere"

_religions, doi:10.3390/rel14121476_

Round 1
Reviewer 1 Report
Comments and Suggestions for Authors
This is a valuable, well-reasoned theological exposition focusing on the tradition of the églises de réveil (awakening churches) in Africa; especially in the Democratic Republic of the Congo. Creative use of the ideas of Ignatius Loyola and H. Richard Niebuhr. Overall, theological assessments are sound. Author provides valuable information on awakening churches and their social impact.
It is primarily a qualitative study -- not an quantitative, empirical study as the author sometimes implies. Need to separate direct observations from theological opinions and operationalize key terms such as : "authentic," "Christ-centered," and so on. Give more concrete examples.
Valuable information on modus operandi of these churches. Fills a gap in the literature. Makes apt comparison of pastors and African politicians. Much of what he reports for African pastors and politicians is true elsewhere.

Reviewer 2 Report
Comments and Suggestions for Authors
Dear author,
Below, please find my suggestions related to your text:
1) in the title, the expression `the angel of light at work` should be placed within quotation marks, because it indicates a methodological lens for evaluating the activity of the pastors of awakening churches.
2) all biblical references should be included after the biblical citations, rather than in the footnotes.
3) the citation style specific to the Religions Journal should be adhered to.
4) I recommend including academic references or empirical data (such as statistics or media-reported cases), in the following situations:
a) when affirming in the text that awakening churches pastors face temptations of materialism, power, or glory, and `many are falling` (rows 183-184)
b) when emphasizing that pastors `preach themselves and put Jesus aside` (rows 195-198)
c) when stating that pastors use religion instrumentally to secure their survival or `writing church estates under their names` (rows 247-248; 252-255)
d) when describing aggressiveness as a effective medium for evangelization in the eglises de reveil (rows 270-279)
e) when referring to the content of the preaching of many pastors of the eglises de reveil (rows 289-291).
This will enhance the assessment of modus operandi of awakening churches.
5) The concept of „radical Christocentrism” is not new in Missiology. It is important to mention relevant contributions on this topic.
6) When you write that Jesus is the model of hope for the Christian people and that `to live a life full of hope means being nourished by an eschatological consciousness (rows 629-638), it seems to me inappropriate to use the words of Rabbi Jonathan Sacks as an argument for or as an exemplification of the idea you support. I believe it would be more suitable for you to cite an author within the Christian tradition to illustrate the idea.
Round 2
Reviewer 1 Report
Comments and Suggestions for Authors
Author has addressed all my concerns. This is an excellent contribution.